# Lessons for Workforce Disaster Planning from the First Nosocomial Outbreak of COVID-19 in Rural Tasmania, Australia: A Case Study

**DOI:** 10.3390/ijerph18157982

**Published:** 2021-07-28

**Authors:** Jessica Hammersley, Carey Mather, Karen Francis

**Affiliations:** 1School of Nursing, College of Health and Medicine, University of Tasmania, Hobart, TAS 7000, Australia; janewett@utas.edu.au; 2School of Nursing, College of Health and Medicine, University of Tasmania, Launceston, TAS 7250, Australia; karen.francis@utas.edu.au

**Keywords:** COVID-19, coronavirus, community, critical discourse analysis, health, rural, nosocomial, workforce, sustainability, communication

## Abstract

The identification and announcement of the COVID-19 pandemic has been a global issue. Disaster preparedness for internal and external threats is inherent within health care environments and requires agile thinking and swift remediation. Nosocomial infection is a risk for recipients of care, especially in hospital settings, which has implications for workforce planning. The aim of this case study was to examine the community response to the internal disaster of the first nosocomial COVID-19 outbreak within an Australian rural health care environment. A critical discourse analysis method was adopted to generate and analyse data collected from three different media platforms during a six-week period. Four main themes were distilled: actions and intent, loss, well-being and recognising choice, and community action. Phase two of the study interrogated these themes to expose the power positioning of speakers and their relationships to the audiences. Strengthening communication with local communities within health care environments must be a priority in any future rural workforce disaster preparedness planning. Maintenance of trust with health service provision and delivery in rural communities is imperative. The inclusion of a robust communication plan within any risk management strategy that meets the needs of the local users of health services is mandatory.

## 1. Introduction

The identification and subsequent announcement of the Coronavirus Disease 2019 (associated with the virus SARS-CoV-2), hereafter known as COVID-19, as a pandemic has been a global issue [1,2]. Preparedness for internal and external disasters is inherent within health care environments, including those in rural areas, and has been widely recognised, particularly in the case study described [3,4,5,6,7]. There is a need for disaster information to be disseminated to the general population and tailored to meet the needs of rural localities, especially when the external threat becomes an internal disaster within the local area [8]. Additionally, strong media communication regarding appropriate health information messaging to prevent misinformation or myth generation reduces anxiety and fear. Media communication encourages communities to respond and make informed decisions to minimise and also to control negative outcomes regarding trust in the health workforce, health care service delivery and the community [8,9,10,11]. The first nosocomial outbreak of COVID-19 in Australia occurred within a rural health care environment. This outbreak is reported as a case study and uses critical discourse analysis as a method for generating, analysing and interpreting the digital and traditional media data during the internal disaster. The use of critical discourse analysis allowed for the voices of the media and its audience to be explored in relation to workforce planning in rural health care settings. Recommendations for disaster planning to mitigate risk to rural workforces are suggested.

### 1.1. Media in the Global Context of COVID-19

In Australia, the advent of COVID-19 resulted in additional funding to support remote digital health service delivery. Access to adequate healthcare was provided to the population whilst reducing the amount of physical contact required and thereby limiting the risk for transmission of COVID-19. In rural areas, remote digital service delivery was impeded by Internet access issues [12]; however, gains were made in using social and other media to maintain communication and support self-isolation [12]. Similarly, the United Kingdom and the United States of America (USA) also utilised digital health services for community education for their health care workers and promotion of communication to citizens [12,13,14,15,16,17,18,19,20]. However, in other countries such as China, the digital media experience regarding communication of COVID-19 information was different to Australia. One study noted that the population generally had lower engagement with government media communication than personal online COVID-19-related posts [1]. Content of government media communication was more likely to include data, information, policies, and official actions, whereas personal posts included empathy, worries, and attributed blame. This study recommended government monitor social media to determine appropriate times to share information and take a more empathetic approach to communication to address public concerns [12,13]. In the USA, a survey of 979 individuals indicated that Internet media information they received about COVID-19 online and awareness of positive cases within their local community were positively associated with an increase in preventative behaviours [14,15]. Internet media has been found to support some groups during the COVID-19 crisis [16]. However, in Turkey, Sevimli [15] described negative media communication during the COVID-19 outbreak. These reports included criticisms of governments for lack of effective action, testing appropriately, providing sufficient personal protective equipment, or providing insufficient information about specific cases [15]. International and Australian research regarding the response to the global pandemic provides insights into how citizens and forms of media communication frame health care information [17]. Access to information and how media framing influences or informs the etic worldview of citizens, and the subsequent translation of messages affects outcomes in local communities [17].

### 1.2. Case Study—Northwest Tasmania, Australia

A COVID-19 outbreak was declared on 3 April 2020 at the North West Regional Hospital (NWRH), a 160-bed facility, on the island state of Tasmania. For this rural community, the sudden escalation of COVID-19 cases sparked the temporary closure of two health facilities and quarantine of all staff and their household members [18]. The period of the COVID-19 outbreak was during the initial period of the Tasmanian experience of the pandemic and occurred during a time when global understanding of how to best manage the virus was changing rapidly [18,19]. Similarly to other parts of the world, this situation created an ethical, moral and physical effect on the community at a rural, state and national level [8,20].

The outbreak at the NWRH occurred on Day 18 of the declared global public health emergency, which was also soon after the implementation of Tasmania’s COVID-19 health response. Three cases of COVID-19 in health care workers were reported between 3 and 4 April 2020, and a further eleven cases were reported over the next three days, resulting in police assistance in contact tracing [18]. Concurrently, ambulance presentations were diverted towards another regional hospital, and patient transfers could only occur with the approval of the Executive Director of Medical Services for the region, while visitors to the two hospitals in the region ceased. A specialist infectious diseases physician was appointed to support the region. On 8 April 2020, the NWRH was escalated to Level 3 of the COVID-19 plan. This internal disaster management included the closure of medical and surgical wards to all new patients, while health care workers were prevented from working across health care facilities within the region [18].

National guidelines regarding the definition of close contact between individuals changed on 9 April 2020 to 15 min cumulative face to face, rather than continuous contact, which required additional contact tracing to be undertaken. Further escalation to Level 4 of the COVID-19 plan occurred, and at this point, all staff were required to self-isolate for a period of fourteen days. On 10 April 2020, ten more cases relating to the outbreak were reported. Households of quarantining hospital staff and all discharged patients, since 27 March 2020, were required to isolate for a period of fourteen days. On 11 April 2020, the State Senior Executive, including the Premier of Tasmania, confirmed the decision to close the NWRH precincts based on the safety and sustainably of service delivery [18]. During this period, approximately 1300 staff and a further 3000–4000 rural household members and close contacts were committed to quarantine for fourteen days. By 21 April 2020, seventy-three staff members, twenty-two patients and nineteen others, including household contacts, were reported to have acquired COVID-19. Through rigorous contact tracing, quarantine and closure of the affected hospitals, the spread of COVID-19 slowed, and the measures proved to be effective in containing the outbreak with emergency and maternity services commencing following a fourteen-day period and other services returning to full operating capacity in the weeks following [18].

### 1.3. Dissemination of Health Information

Online health information has increasingly become available to populations worldwide [13,16]. Individuals search for health information and also become exposed to it incidentally whilst engaged in other activities. Access to information not only informs populations, but it also influences health decision making, perception of risk, and engagement in preventative behaviours [14,15]. Social and other media have and continue to play a significant role in the dissemination of information that influences the public. The Tasmanian Department of Health acknowledged that during the NWRH situation, concerns were raised by the health workforce that social and other media was a better source of information than their employer [21]. This situation was to some extent due to the need to maintain the privacy of affected persons and difficulty communicating through official channels outside normal working hours [18].

The aim of this research was to analyse the first nosocomial outbreak of COVID-19 in Australia, which occurred within a rural health care environment on the island of Tasmania, Australia. As a result of this outbreak, there are implications for preparing rural workforces for future internal disasters. For this research, the nosocomial outbreak is reported as a case study [22]. Critical discourse analysis is used as the method for generating, analysing and interpreting the digital and traditional media data generated over a six-week period during this internal disaster.

## 2. Methodology and Methods

A case study methodology can be employed when a detailed understanding of an issue (the impact of COVID-19) that occurs in a real-life context is the focus of research [23]. This case is bounded geographically to the Northwest Coast of Tasmania and discourses emanating from government, traditional and social media were the targeted sources of data. Critical Discourse Analysis (CDA) was chosen as an appropriate method for generation, analysis and interpretation of these data. CDA supports the interrogation of written (textual) or spoken language or non-verbal communication, visual images or multimedia that occur within a social and cultural context [24,25]. Language is a powerful tool that creates opportunities for influencing the values and beliefs of others. Recognising why language and how language is used to achieve predetermined and causal outcomes are core intentions of DA [25]. Adopting a critical discourse analysis (CDA) approach focusses the researcher’s attention on ideological positioning of the speaker/s, power relationships, methods of manipulation that are used to influence others either overtly or covertly, exploitation, and the structures that enable inequalities to be established and maintained [25,26].

Mullet [25] provides a seven-stage framework for completing a CDA:Select the discourse;Locate and prepare data sources;Explore the background of each text;Code texts and identify overarching themes;Analyse the external relations in the texts (interdiscursivity);Analyse the internal relations in the texts;Interpret the data (Table 1).

## 3. Results

Two phases of analysis were undertaken to expose overt and covert discourses within the texts examined. The first phase’s findings were framed and evolved from the CDA of original data, which were collected over a six-week period. Two researchers analysed this data by finding commonalities and differences within the data sets, which resulted in four key themes: Action and Intent, Loss, Wellbeing and Recognising Choice, and Community Action. These themes are linked to community experience, government and community actions, and the information portrayed through various media platforms. Within each of these categories, there is a strong sense of the need to recognise the voices who were speaking and how the chosen audience perceived the information delivered. In the second phase of analysis, data were interrogated by the researchers to expose who was speaking and why, and the power positioning of speakers along with their relationships to the audiences listening. The results are presented within each of these phases.

### 3.1. Phase One

#### 3.1.1. Discourse Media Data Context

##### Print Media

The *Advocate* newspaper is the regional newspaper of Northwest Tasmania. It is currently owned by Antony Catalino of Domain publishing and the Thorney Investment Group, which have a 50% share each [28]. Domain and Thorney purchased the newspaper from the Regional Publishing Group in 2019 and currently have a press sharing arrangement with Fairfax Media and News Corporation. The *Examiner* and *Mercury* newspapers are owned by News Corporation. The *Mercury* newspaper serves the community of Southern Tasmania, while the *Examiner* is the preferred newspaper in the north of the State of Tasmania.

##### Social Media

Social media platform Instagram was purchased by Facebook in 2012 by the current chief executive officer, Mark Zuckerburg. Instagram places community guidelines on its users that encourage an “authentic and safe place for inspiration and expression” [31]. Due to the recent global COVID-19 pandemic, Instagram has ensured engagement with experts to ensure that information shared is accurate and appropriate to keep members of the community safe [31].

##### Internet Search Engines

Alphabet Inc is the parent company of Google and is owned by five major shareholders. The shareholders are the two co-founders, Vanguard Group Inc, BlackRock Inc and T Rowe Price Associates Inc. The newspapers are curated by journalists retrieving information from other journalists and fashioning it for their community readership. Direct messages from citizens or grassroots groups are less curated, although they are subject to “cleaning” via third party companies with agendas and biases of their own. Search engines are also curated, depending on the user’s previous searches [32].

#### 3.1.2. Key Themes

##### Actions and Intent

The theme of actions and intent arose primarily from the discourse analysed in print media. Open and axial coding of key headlines related to COVID-19 identified repetitive forms of punitive messaging in COVID-19-related media headlines. Frequent terms associated with the theme of actions and intent were observed in the following headlines: authority, fines, investigation, police, prisoners, mayors, handcuffs, consequence, crisis, ground zero, fight, bad luck, gloves are off, lucky to be alive, crises and lockdown extensions. Although at times the use of the terms and references outlined were framed otherwise, the use of the key terms can be perceived as and allude to the power differentials in social structure, the degree of control of action by local authorities and organisation, influencing the cognition and actions of the public. The analysis of each point of data collection reflected punitive messages and was associated with status, power and control-laden relationships with authority/authorities. This messaging influenced the provision of lockdown law(s) and the consumer readership within the general public. A sense of increased surveillance and suspicion of community members activities and reprimand for those who chose to be defiant and offend were noted at the beginning of the six-week point of data collection. Local newspaper headlines such as “helicopters to police compliance with COVID-19 restrictions in Tasmania” provides an explicit example of this surveillance [33]. Rule and ban breaking are examples of negative language used to shift the discourse from that of a health pandemic to provoking victim-blaming alternatives. 

The shifting of the narrative was also noted to draw associations with stigmatising lifestyle behaviours and the notion of criminality with direct references to the additional COVID-19-related penalties for speeding drivers, drug users and party goers. Thought-provoking punitive headlines further emphasised risks associated with lockdown extensions as well as questioning of health workforce integrity with local newspapers, indicating that over “60 staff were stood down as coronavirus outbreak investigation continues” [34]. This positioning of power over health care workers at the centre of the COVID-19 outbreak left community members feeling a sense of helplessness and blame for their community. With the temporary closure of a health care service and enforced periods of quarantine for health care workers and lockdown for the region, the disruption to health care access and workforce was evident through the discourse. Due to the language and messages being portrayed within the local media, health care workers perceived they were blamed for playing a role in the outbreak.

##### Loss

An emphasis of this CDA was placed on the concept of loss. The sense of loss identified throughout the discourse was strongly linked to the rights of civilians and the loss of trust in essential services as a focal point within local print media and via Internet search engines such as Google. Messaging from these platforms hinted at the use of restrictions being placed on individuals due to COVID-19 as a violation of civil, sociocultural and political rights. Loss of trust within the Tasmanian health care system and health care workers employed was emphasised within the data. With the enforcement of quarantine for approximately 1300 staff and Level 4 of the COVID-19 plan in place, the discourse portrayed a sense of loss of identity and purpose for health care workers within the region. Newspaper headlines insinuated that health care staff were stood down due to poor behaviours, rather than as a necessary step to reduce further nosocomial infections within the hospital and community [34]. These headlines implied a loss of trust in health care workers who provided direct health care service delivery within the rural community. 

Within the discourse, the loss of freedoms that individuals within rural communities experienced during periods of hard lockdown was identified through the closure of businesses and education systems. The loss of individual rights was evident in the use of harsh punitive penalties described in theme one, including lengthened periods of restrictions and the use of population surveillance. Headlines such as “Gloves are off in Coronavirus fight” [35] described the use of patrols and law enforcement to suppress those seen as breaking rules and risking the safety of others. Periods of lockdown and the closure of intrastate borders meant that individuals were confined to their local government area, resulting in social isolation and restricting individuals’ abilities to participate in employment, education, health, and industry. The loss of business grossly impacted small communities with police “monitoring Tasmanian retailers” to ensure compliance of social distancing requirements, reducing numbers of patrons in stores [36]. Through the six-week period of data collection, Jarvie [36] described the extension of retail restrictions for the Northwest population throughout a period of lockdown as a direct influence of community behaviour. The language used led to community concerns about the economy, jobs and the wellbeing of individuals. Furthermore, print media portrayed these restrictions as a reflection of poor community behaviour, rather than creating a sense of awareness of means to provide safe health care environments for the rural Tasmanian community.

##### Wellbeing and Recognising Choice

It was identified throughout the data collection period that information delivered via social media platforms such as Instagram had a strong sense of encouraging the mental and physical wellbeing of their followers through changed ways of living and optimism. Individuals posting on their own accounts and those posting on behalf of local businesses and sporting organisations provided their audiences with information on self-care practices that met social distancing guidelines. Alternative wellbeing practices that could be undertaken throughout their period of isolation were also suggested. These posts and images captured activities such as cooking, arts, reading, exploring nature in our own backyards and spending time with friends and family. This messaging allowed for the appreciation of activities or objects that were made available to individuals in a time when a waver in their beliefs and hope was indicated. These posts also provided individuals with different adaptations with respect to how they could remain connected by undertaking self-care practices in social isolation when physical contact was not permitted. Discourse data showed different means of adapting technology for maintaining social connections and individuals’ livelihood such as Apple FaceTime to speak with family and friends, live streaming of concerts and shows, and the use of Zoom videoconferencing platform. 

Posts from sporting and fitness organisations included inspiring images of community members maintaining a level of fitness by undertaking their normal gym workouts from the comfort of their homes. Such material extended a sense of motivation and appreciation for the mental and physical health that they were able to maintain during a time when Stage 3 restrictions were imposed. Within the discourse, it was recognised that providing individuals with an opportunity to spark reflection on past events led to aspirations for their future. The audiences of these followers provided support in the making of decisions as a response to recognising the need for change in how they chose to live moving forward following the global pandemic.

##### Community Action

Key data representative of community action was centred around the collective experiences of COVID-19-related precautions and practices in response to public health advice. This response was observed across all forms of media during the period of data collection, aligning with the peak of local community transmission. Frequently observed terms and phrases related to this theme featured in headlines, Internet search engines and hashtags. These communications included references to social and physical distancing, isolation, quarantine, staying at home and restrictions. A strong sense of continuing community spirit was a particularly sustained feature of the Instagram platform. Data collection via the Instagram platform was obtained by searching regions and locations to capture local data. A cluster of posts with a strong sense of sharing experiences of COVID-19-related practices, reinforcing standards and quality of such practices and support for self and others in this community action was observed. This theme took shape in a variety of pictorial postings aligning with key terms such as text memes, digital impressions reinforcing physical distancing of 1.5 m, handwashing and mask-wearing and the adoption of individual self-care activities. 

For health care workers, the sense of community was evident through local citizens coming together online to celebrate and praise the efforts of staff in other regional hospitals for their work during the period of Stage 4 COVID-19 restrictions. While their praise of health care workers in one region was recognised, fear was instilled amongst the health care workforce and community within the Northwest region secondary to Internet searches of local government information [37]. Statements such as “Epicentres of infection” and “Misconduct of people impacts on disease transmission” resulted in mistrust for local health care workers during a period where their workforce was disbanded as a result of the nosocomial infection and internal threat within local health care environments [37].

Social connectivity during the time of physical distancing was a repetitive nuance in the discourse framing on the Instagram platform, as was the vigilance pertaining to mental health implications for self and others in the face of a collective experience of the loss of liberty and freedom. This sentiment was expressed by posting well-being reminders and strategies along with a critical consciousness of checking in with others, elders and loved ones. Expressed concern for inclusivity with those who may be perceived as having the greatest risk concerning the influence of COVID-19-related social isolation, rather than the COVID-19 virus itself, was observed with reference to the communities of lesbian, gay, bisexual, transgender, queer and intersex (LGBTQI) individuals. The notion of gratitude for health and safety of self and others and reflecting on previous times of full expression of individual and collective freedom was portrayed. Holidaying in the past and plans for when the future normativity returns were specifically observed, along with forms of explicit goal-setting, past events of collective nature and posts targeted at motivating others also contributed to a sense of community action in the time of a pandemic. 

The Instagram platform, more so than the other forms of media analysed, offered valuable insight into perspectives and experiences of the general public. Posts from smaller-scale local businesses of rural communities offered clarity about the day-to-day realities of living in social isolation. The new challenges imposed by the pandemic for already geographically isolated communities within the island state were often superseded by community actions of resilience. Examples of community resilience were observed in the sharing of practical, pragmatic innovations in day-to-day living, eating, exercise, working and communication. Resilience in the adaption to the processes of the employed, working from home, was observed as an expression of enthusiasm, pride and accomplishment, as well as taking recognition for exceeding expectations in the reform efforts taken. 

A shift in paradigm, productivity and responding to unmet community needs was acknowledged, as well as the point of community connectivity, with measures related to prompt re-design and extension of services such as digital media health service provision. New initiatives were credited to the performance of community organisations, smaller-scale businesses and sole operators. The data collection coincided with the community significance of Earth Day and Mother’s Day with evidence of discourse promoting a sense of community spirit and survivorship shaped by the refuge and gratitude the celebrations offered individuals but also offering businesses the opportunity to harness gains with material advertising.

### 3.2. Phase Two

Data collected were interrogated to expose who was speaking and why, the power positioning of speakers, and their relationships to the audiences listening [24,38]. Within Mullet’s [25] CDA framework, exploring the background of each text, including the overall slant, intended audience, intended purpose of the text, publisher and writer characteristics, allowed for the examination of the media and to help understand the true power positioning of the speaker(s) [24,25]. Within print media and Internet search engines, government voices were pervasive [24]. The texts provided by these sources provided information that was directional, underpinned by ethical and moral rationalisation and threatened punishment if compliance was not achieved. Media provided a strong option for a political lens to be used, and audiences of these media were compelled to do “the right thing’, a moral obligation, or be punished if they failed. This action was justified on the grounds that non-compliance resulted in preventable harm to others. This notion was seen particularly within local newspaper print, which included headlines such as “Speeding drivers, drug users, party goers cop extra COVID-19 fines” [39]. Print media portrayed health care workers in a negative way with headlines such as “60 staff stood down” [34]. However, these staff were placed in self-isolation for a period of two weeks and were then permitted to return to work. The Australian Government Fair Work Ombudsman [40] stated that employers cannot stand down employees because of a deterioration in business conditions or if they have COVID-19. Therefore, the terminology used within print media was incorrect, potentially portraying misinformation to audiences and promoting a view of lack of trust in the local health care workforce.

Within online search engines such as Google, algorithms are used to direct “preferred” content to the searcher. Content is paid, sponsored or pushed; thus identifying “biased’ views is markedly increased. Trending can occur through targeted action by groups and individuals to serve their own purposes and to increase visibility. No media communication is without bias, and it depends on how overt/covert the editorial control is in the print media and the direction of the chief executive officers of the Internet social media and search engine companies. Within the media, the voice appears divided. On one hand, messaging reflected that of the national and local government, and on the other, the government appears “attacked”. The power base from which the media speak is both supportive and confrontational [41]. There is a challenge presented to the Australian and Tasmanian government to be accountable through greater transparency; however, the challenge is how this accountability is represented within media communication.

#### 3.2.1. Social Media

Social media presented a different lens of living through a global pandemic in rural Tasmania. Instagram is a platform for individual expression, and, similar to other forms of social media, enables individuals to follow others they are interested in through the context of sponsored and unsponsored posting activity. The social media platform of Instagram offered insight into the experiences of local community members directly impacted by the hard lockdown and public health measures. The information available to consumers within the realm of social media generally provided content that was underrepresented in other forms of media data sets. It reflected individual community members’ experiences including success and hardship in the context of wellbeing, COVID-19 restrictions and supporting one another, as well as the broader local community, local businesses, local arts sector, sports and recreation organisations and education. The Instagram authorship for this data collection tended to represent community members who had less official or authoritarian positions to gain influence in other forms of media. The platform enabled community members to position themselves as political activists by content contributions that supported the metacognitions of others during a global pandemic by asking questions, provoking action and providing critique of others and those instigated by local government and public health officials.

During the period of data collection, there was the addition of content with a continued government presence superseding the platform sponsorship. This content was observed as a direct link to the government COVID-19 website visualised at the top of the consumer’s viewing feed. The hierarchical presence of this link provided a constant visual cue of current times and a direct access point to valid and reliable current public health information for platform consumers to access. Another unique aspect of the findings from the content from this media platform was the trending of local e-commerce that leveraged off local experiences during the time of COVID-19 using perspectives of local businesses and perceived consumer needs as a point of engagement. This media platform tended to be more supportive of consumer needs during the data collection period compared to other media platforms described while ensuring an authentic and safe place for inspiration and expression by its users. Due to restrictions on health care workers and the use of social media to express their feelings and thoughts as a member of a health care organisation, it was difficult to determine the influence on the workforce through a social media lens. However, praise for health care workers within the state was evident and the messaging to “stay home” was emphasised to keep health care workers and their patients safe during periods of the Level 3 and 4 COVID-19 plan.

#### 3.2.2. Who Is Listening?

Media communication is part of everyday life for many Australians, and how they interact with it is based on the generation and the media form [42]. A media audience can be described as an individual or group of individuals who consume media and are considered either active or passive [42]. An active audience consumes media by engaging with the text, not accepting every media message, questions what they observe, and develop their own interpretation of what is presented to them based on their life experiences, education, and family and cultural influences [42]. A passive audience is considered one that merely observes an event rather than responding to it [42]. The audiences of media are based on personal preference; however, the relative influence on personal experience, cultural background, educational attainment, work colleagues and the perceived Australian “youth culture” may play a significant role in the attitudes that control this preference [42].

In relation to the media considered, print media such as local newspapers analysed showed a reach to 101,000 Tasmanians in a 12-month period (*The Mercury*—61,000; The *Examiner*—30,000; and *The Advocate*—20,000) [43]. For social media platform Instagram, in October 2020, Australia accounted for 41.3% of Instagram’s entire population (10,620,000 people) [44]. People aged between 25–34 were the largest user group (3,200,000 people), with women being the majority of this group (56.5%) [44]. Internet search engines such as Google are used significantly throughout the world to source information related to health services and information. For Australians interacting with search engine platforms, 92.41% of all search engine use between March 2020–2021 was linked to the Google search engine [45]. This is significant in comparison to the use of other platforms such as Bing, Yahoo, Baidu and YANDEX, which received less than 2.46% of all use in Australia [45]. Of the 25,693,000 people residing in Australia, 541,000 reside within Tasmania [46]. While Google accounted for the highest search engine usage within Australia, there are currently no available statistics that link the use of the Google search engine to any given state within Australia.

## 4. Discussion

Foucault [47] argued that power is everywhere and comes from everywhere. Knowledge, he argued, is power and is established, maintained and accepted by society, while the accepted truths can be contested and new truths negotiated [47]. This study has shown that three competing discourses were able to establish a power base from which to speak. The government voice claimed attention as the established “truth”, and the print media utilised power by positioning as both an ally of the government and protagonist, speaking on behalf of society [48]. In contrast, social media speakers claimed the right to speak as members of society forcing a new “societal truth” to be negotiated [41]. With rapid increases in the number of Internet users, social media plays a prominent role in modern society [49]. Von Nordheim [50] describes the accessibility of social media as a pathway to news as very popular; however, the information may present questionable accuracy. Lin et al. [49] note that the dissemination of inaccurate information was previously thought to have potentially detrimental effects on communities; however, their study suggests that favourable psycho-behavioural responses can be attributed to information viewed via social media. This is highlighted within the themes of Wellbeing and Recognising Choice and Community Action, whereby social media users were provided with social connectivity in a time of social distancing and periods of isolation and loss. This was particularly extended to those employed as essential workers such as the local health workforce and those making up vulnerable groups such as the LGBTQI, elderly and disadvantaged youth.

Headlines within print media are considered significant; moreover, Le [51] described the short timeframe allocated to current newspaper reading as “heightening” the value of a headline. O’Donnell et al. [52] stated that Australian readers spend approximately 30 min or less reading a newspaper. Given the short timeframe that audiences engage in print media, major headlines now determine whether the readership engages further with the media based on the attractiveness of the headline [51]. Holsanova et al. [53] suggest that headlines, followed by pictures, are the audiences’ first point of entry to further reading. Within the themes of Actions and Intent and Loss, the headline “60 staff stood down” was highlighted with an accompanying image of an abandoned NWRH [34]. The terminology used resulted in a potential negative portrayal of the health care workforce within the Northwest region. While Kovach and Rosenstiel [54] describe the social purpose of journalism as providing people with information “they need to be free and self-governing” (p. 12), this notion was not evident within the print media and Internet search engine findings of this study. However, Christians et al. [55] describe the use of media as a way of “covering crises and campaigns [and] acts as a check on power by alerting citizens to problems” (p. 97).

The theme of Actions and Intent in the media was not unusual, with many forms of Western media using headlines that relate to punitive actions to entice their readership [51]. Discussion about the different perspectives of punitive or assistive measures in ensuring community engagement and how acceptable these are in different situations have been raised [56]. One such discussion focused on the emic perception that punitive responses were seen to be more acceptable “within” the group or culture than the etic perspective, which has been attributed to the greater influence that other group members have on the safety of the group [56]. White and Delacroix [57] confirmed that punitive media communication is viewed from a less reliable blame culture rather than a reliable and just culture. As a result, a blame culture is likely to have a greater impact on vulnerable groups within the population and can be closely linked to those living within rural communities [58]. Additionally, increased inclusivity and critical consciousness of individuals could be raised if a whole-community approach is genuinely applied, leading to increases in well-being and community-focused actions. Furthermore, a communication plan that is inclusive of the community could reduce the feelings of loss and engender the creation of resilience, enabling greater preparedness for any future disasters. The current Tasmanian Emergency Management Plan [59] outlined communication strategies for external disasters to inform members of Tasmanian communities. Means such as social media, emergency alert, television “screen crawls”, web-based information, call centre services, community meetings, community-based information centres and messaging in multiple languages as and when required have been highlighted within the current Tasmanian Emergency Management Plan [59]. However, this research demonstrates local community feedback as being vital to ensure messages are appropriate, transparent and reliable. Furthermore, this case study highlights negative health care workforce and community perceptions when disaster preparation does not include a robust bi-directional communication plan and support by the print media.

The COVID-19 outbreak in rural Tasmania is a unique case study, as it was the first nosocomial hospital outbreak of COVID-19 in Australia. The Tasmanian State Government Department of Health invested substantial resources in informing the whole community about what was occurring within the local area and how they should respond. This information was supported by the Australian government, which concurrently was distributing information across the country at a national level. A later independent review into the nosocomial hospital outbreak at the NWRH identified several challenges to this communication strategy, including the ability for people to source their own information and the frequent changes in guidelines, which undermined the trust of the community [19]. Although the community was aware of what was occurring globally through their viewing of the media as described in Table 1, the rapid pace of change and mixed media messaging provided lessons for future workforce planning. Planning requires considering the sustainability of the workforce and ensuring that any future disaster management plan includes a robust multichannel communication strategy to promote positive community action and health outcomes and supports the health workforce that was subjected to internal and external disaster management. This disaster management incorporated government actions to reduce the risk of further infection, which included prohibiting the health care workforce from practicing across multiple health care facilities [7]. Additionally, planning for internal and external disasters within any rural health care environment is imperative [59]. The United Nations Office for Disaster Risk Reduction [60] describe the occurrence of such internal disasters as unpredictable, resulting in chaos and potential mass casualties. Long-term effects on health care systems such as social, physical, psychological, environmental and economic effects need to be monitored and included in planning. Risk mitigation for maintaining the sustainability of the local health care workforce during internal disasters is vital [61]. Prevention of a similar situation of the media questioning the behaviour of the health care workers and the implications of lack of trust of the health workforce experienced in this case study could be averted if media correctly proposed headlines to reflect accuracy rather than portray the health workforce negatively [53,55]. Additionally, the themes of Community Action, Actions and Intent and Loss further reflect the community anxiety and fear created by the print media. However, there are lessons to be considered to promote community cohesion if or when a future internal disaster occurs within a rural health care environment.

### 4.1. Recommendations for the Future

Phase one and two of this CDA provide lessons for future workforce disaster planning. Internal disaster preparedness needs to include media communication strategies that suit local environments. Transparent actions by local government and affected health care organisations are essential for maintaining trust in health care workers and service provision by members of the community. Furthermore, a tailored strategy for each local community enables seeking out vulnerable groups such as the LGBTQI, elderly and disadvantaged youth to support them with appropriate messaging, prevent misinformation or myth generation and reduce anxiety or fear [4].

The communication plan needs to lever a multichannel approach and consider local media nuances and evolving digital platforms. Whilst internal disaster requires swift mitigation, it is imperative that media communication is bi-directional between the staff of health care environments and the community. Feedback from the community will facilitate appropriate messaging. The delivery of consistent, cohesive messaging to the health care workforce and members of the community requires support from appropriate trusted personnel. Additionally, an internal communication plan is vital to ensure that the health care workforce is informed. The health care workforce is an integral component of risk management of an internal disaster, as they can be harnessed at an individual level to promote the dissemination of appropriate information, prevent misinformation and dispel myths through personal media channels when required or asked. Factors such as presenteeism need to be considered and support provided for the health care workforce in creating an environment that represents health care workers accurately [62]. Disaster-preparation training programs require redesign to incorporate the advantages of digital technology. The speed of transmission of COVID-19 shows that health care environments need to be agile and swift in response to internal and external disasters, especially in rural areas, which may be less prepared than tertiary facilities [7,18].

### 4.2. Strengths and Limitations

As with any study, there are limitations to this study. The data collected occurred at fortnightly intervals over a six-week period from three different forms of media. While the period of data collection did not extend beyond the six-week period, it can be considered a strength of the study that the researchers collected and analysed data from three different forms of media during the disaster. While the generalisability of these findings is limited by its primary intention to provide information in a local context for the Tasmania health service and community, understandings from this study can inform other rural health care services within Australia and globally. As this was the first nosocomial outbreak of COVID-19 in Australia, this study highlights community influence as a key component of internal workforce disaster preparedness. Although Mullet’s [25] CDA framework was used to guide the researchers, there is potential for unconscious bias. However, the framework (Table 1) that was adhered to during data collection provides evidence of rigour.

## 5. Conclusions

Collaboration of community and health care environments to reduce vulnerabilities that emerge during disasters remains vital. Using the method of CDA, this case study highlighted that the COVID-19 pandemic has demonstrated external threats as invoking internal disasters that can adversely affect the health and wellbeing of the health care workforce and the local community. This study found that government messaging that embraced a unidirectional and castigatory approach may have alienated members of the community, while traditional print media utilised its position of power to support the government, and also acted as a protagonist ensuring a delicate balance of meeting competing needs: government and the community. Social media conversely enabled community members to articulate their needs, ask questions, receive feedback, and garner support for causes they deemed important. Issues of importance identified by community members, however, were not necessarily aligned to those advocated by the government or the traditional print media.

Due to the ease of accessibility, social media play a significant role in the daily lives of many Australians, while print media and Internet search engines provide curative and mixed messaging of important health information. Appropriate and salient messaging can avert negative community perceptions of action and intent or loss while also promoting positive community action and ensuring the community trust in health care services remains. While the threat of COVID-19 is currently remnant in a global sense, the influence that media communication places on rural communities and those working within health care services and delivery of care remains significant.

## Figures and Tables

**Table 1 ijerph-18-07982-t001:** Framework of Mullet’s CDA [25] applied to this case study.

Stage of Analysis:	Description:	Focus and Actions:
Select the discourse	Select a discourse related to injustice or inequity in society.	What is the community impact on the health and wellbeing of a rural COVID-19 pandemic in Tasmania?
2. Locate and prepare data sources	Select data sources (media) and prepare the data for analysis.	A method of data collection triangulation was utilised, with three researchers independently collecting primary data from three different media platforms, across three separate points of fortnightly dates over a six-week period (9 April, 23 April, and 7 May 2020). An analysis of publicly available information occurred on each platform for a 24 h period or on top search results on the dates selected. Data from each media platform were collected by three individual researchers and was audited on the following content: - Who was speaking? - Why are they speaking? - Who allowed the speaking? - Who was listening (audience)? **1.** **Internet Search Engine** A Google search was undertaken using the words “Tasmania, rural, COVID-19”. The top search results were collected for analysis. Top search results were generated from organisations such as Tourism Tasmania, Tasmania State Government and Worksafe Tasmania. **2.** **Print Media** Three state local newspapers were reviewed on each of the data collection dates. These included *The Advocate*, *Examiner* and *Mercury*. Articles from these newspapers were selected and reviewed based on headlines and keywords they contained. Each article selected featured COVID-19 as its focus, for example, “Helicopters to police compliance with COVID-19 restrictions in Tasmania”. **3.** **Social Media** The social media platform Instagram was used to obtain data on each date. To gather these data, the researcher used a combination of hashtags such as #Tasmania, #northwestcoast, #Burnie, #Queenstown, #rural, #Cygnet, #COVID-19 to generate data. The geographical tagging of Tasmanian suburbs was also utilised. Each publicly available photo or video was reviewed and selected based on its inclusion of COVID-19-related content. For example, “self-care, keeping well in isolation, social distancing, community support”.
3. Explore the background of each text	Examine the social and historical context of the media.	Google is an American multinational technology company that specialises in internet-related services and products, primarily featuring a search engine. Google search was founded over two decades ago by Larry Page and Sergey Brin. Currently, there are several top shareholders including Vanguard and the original founders [27]. The *Advocate* and *Examiner* newspapers sourced for this discourse analysis are owned by Australian Community Media and Printing (ACM). This Australian media company is responsible for over 160 regional publications [28]. The *Mercury* newspaper sourced for this CDA is owned by News Corp Australia. This Australian media company is one of the largest conglomerates with approximately 142 newspapers (102 suburban). Their group spans across print, Internet and subscription television [29]. The social media platform Instagram is an American photo and video sharing social networking service owned by Facebook and was originally launched in October 2010. The app allows its audience to upload media that can be organised by hashtags and geographical tagging. Posts can be shared publicly or with pre-approved followers of users [30].
4. Code texts and identify overarching themes	Identify the major themes and subthemes using the choice of qualitative coding methods.	Investigator triangulation was utilised. Initial open coding was determined by each of the researchers undertaking data collection independently. Utilising a process of inductive coding, two researchers then compiled the initial codes from all sources individually, where they identified emerging themes and ideas from each data set according to the researcher. Axial coding was then used for each data set, and four main emerging themes were determined.
5. Analyse the external relations in the texts	Examine social relations that control the production of the media.	Google is an accessible search tool that uses algorithms that allow organisations to share their product content with their users. The use of algorithms empowers media with a dominant voice to reach Australians in all regions, resulting in information with political agendas being more prominent in search results. The local newspapers identified for this CDA are all owned by large Australian media corporations. The lens in which they portray information may be viewed by their own political agenda(s) and interests. The social media platform Instagram has community guidelines that encourage an “authentic and safe place for inspiration and expression” (Instagram, 2020). Due to the recent global COVID-19 pandemic, Instagram has engaged with experts to ensure that information shared is accurate and appropriate to keep members of the community safe [31].
6. Analyse the internal relations in the texts	Examine the language for indications of the aims of media.	Headlines, leading statements, images and high-frequency keywords that identified the influence of COVID-19 on health and wellbeing of rural communities in Tasmania were examined. Statements from Google and local newspaper data sets such as “bad luck, lockdown extended, health care workers stood down, investigations continue” imply that they are expressing the truth and omit details. In contrast, keywords and photos including “self-care, nature, exercise, food and self-sufficiency” were highlighted on Instagram, implying unitary truth and enables inequalities to have a voice.
7. Interpret the data	Interpret the meanings of the major themes, external relations and internal relations identified.	Four main themes were formed through the analysis of data collected: - Actions and Intent - Loss - Well-being and Choice - Community Action

## Data Availability

Data available on request from the corresponding author.

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
