# Peer review of "Lessons for Workforce Disaster Planning from the First Nosocomial Outbreak of COVID-19 in Rural Tasmania, Australia: A Case Study"

_ijerph, 2021, doi:10.3390/ijerph18157982_

Round 1

Reviewer 1 Report

This version has been significantly improved; all previous concerns have been satisfactorily addressed

This manuscript is a resubmission of an earlier submission. The following is a list of the peer review reports and author responses from that submission.

Round 1

Reviewer 1 Report

Firstly, I declare that I’m not a qualitative researcher. I have a strong interest in rural health and the rural workforce, hence my interest in the paper.

Overall, I believe that the authors have done a solid job with their research methods, moderate presentation of results, stronger discussion of the findings and implications (though someone of more expertise in the DA method will need to confirm).

However, I found the front end of the paper to be rather weak in comparison, notably sections 1-4.

  • The Introduction (Sect 1) is somewhat written without direction; it doesn’t set up the paper in any manner…in particular, there is no connection to Sect 2-4
  • Section 2 is largely irrelevant (after the Intro), but before any context to the study is revealed. The authors make a highly speculative (inappropriate) statement in the last sentence of Sec 2. This may be appropriate in the Conclusion, but not in the 2nd paragraph
  • Section 3 is >80% unrelated to ‘rurality’, notably all of para 2 and most of para 3. The Introduction to discourse at the end of this Section comes wholly unexpectedly, it doesn’t belong in that heading.
  • Section 4 doesn’t make sense to have it separately to Section 5 (and perhaps Section 2).

In para 1, sect 5 there is a typo – they have “permitted from” but they mean the opposite, something like “prevented from”.

Again, Section 6 does not fit the flow of the paper. It doesn’t connect with Section 5 (it needs to, otherwise why focus the study on this). Neither does it setup the chosen methodology. Critically, no aim of the study is given (other than in the Abstract).

As noted, I have little expertise in discourse analysis. However, I was left wondering why section 8 is required. I suspect it is important to contextualizing the DA results, but its relevance is not stated.

Across Sections 10-11 (the results), these are presented in some very long and hard to read paragraphs. E.g. 10.1 is nearly 35 lines, 10.2 about 30 lines. There are many changes of direction within these paragraphs. One example - Within the ‘loss’ one, a critical point to the overall paper of ‘loss of trust’ (of the health professionals) is hidden…and is a very different point to loss of freedom, etc.

Under Phase 2, they state the intent is to expose who was speaking (and why), but I didn’t find it achieved this well. Notably, I didn’t understand why this seemed to be wholly disconnected from the 4 key themes (even though Section 9 suggests it will be).

Minor point – Section 12 is missing.

I felt that the Discussion section was stronger, it did start to bring together Sections 10 and 11 (which I expected in Sect 11) AND it then discussed implications of those findings.

However, I felt that the Conclusion was too focused on the role of media, straying too far from the aim (as per the Abstract) of the actual community response (rather than ‘portrayal’, etc)

Reviewer 2 Report

This was a very interesting manuscript to read.  It was generally well written.  I would suggest the following changes:

Abstract - There is no mention of a case study in the abstract and the first three sentences lend me to think that the paper would be able how to reduce nosocomial infections in hospitals.  

Rural Tasmanian Workforce - I could not see why this section was focused on nurses and suspect it is because of the authors background however it does not have specific relevance to the study and could be removed.  

3. Rurality in the global context of COVID-19 - what is the 'national health service'?  The Australian health service is much more complex than this however suggests that it is similar to health systems in other countries which is it not.  Telehealth services was mentioned however this was not mentioned in the results or discussion.  This section seems to pick out specific countries and it was not evident why.

In Table 1, there is reference to 'nurse researchers'.  Why not just 'researchers' given that research conducted by researchers from other specific disciplines is not referenced by their discipline.

Future Directions - in the first sentence, suggest that it is information rather than 'lessons' for the future workforce disaster planning.

There were very long paragraphs results and discussion sections which should be divided into several paragraphs.

Use of capitals was inconsistent eg. google.